# Precision Agriculture in Brazil: The Trajectory of 25 Years of Scientific Research

Maurício Roberto Cherubin [1],*, Júnior Melo Damian [1], Tiago Rodrigues Tavares [2],*, Rodrigo Gonçalves Trevisan [3], André Freitas Colaço [4], Mateus Tonini Eitelwein [3], Maurício Martello [5], Ricardo Yassushi Inamasu [6], Osmar Henrique de Castro Pias [7] and José Paulo Molin [5]

1    Department of Soil Science, "Luiz de Queiroz" College of Agriculture (ESALQ), University of São Paulo (USP), Piracicaba 13418-900, Brazil
2    Center of Nuclear Energy in Agriculture (CENA), University of Sao Paulo (USP), Piracicaba 13416-000, Brazil
3    Smart Sensing Brasil, Piracicaba 13416-404, Brazil
4    CSIRO, Waite Campus, Locked Bag 2, Glen Osmond 5064, Australia
5    Department of Biosystems Engineering, "Luiz de Queiroz" College of Agriculture (ESALQ), University of São Paulo (USP), Piracicaba 13418-900, Brazil
6    Embrapa Instrumentação, Brazilian Agricultural Research Corporation, Sao Carlos 13560-970, Brazil
7    Federal Institute of Education, Science and Technology Farroupilha, Júlio de Castilhos 98130-000, Brazil
*    Correspondence: cherubin@usp.br (M.R.C.); tiagosrt@usp.br (T.R.T.)

**Abstract:** Precision agriculture (PA) stands out as an innovative way to manage production resources, increasing the efficiency and the socioeconomic and environmental sustainability of agricultural systems. In Brazil, the principles and tools of PA started to be adopted in the late 1990s. To reveal the scientific trajectory and advances in PA taken over the past 25 years in Brazil, we conducted a comprehensive and systematic literature review. After searching for available peer-reviewed literature, 442 publications were selected to compose the database. Our bibliometric review showed that the scientific PA network is growing in Brazil, with the number and quality of publications, the number of interactions among research groups, and the number of international collaborations increasing. Soil and plant management are the two main pillars of PA research (~61% of the publications). More recently, research has evolved to include other areas, such as the use of proximal sensors to monitor soil and crop development, remote sensing using images from satellites and remotely piloted aircraft systems, and the development of decision support tools. A substantial part of Brazilian PA research is marked by the evaluation and adaptation of imported technologies, a scenario that is slowly changing with the growth of well-trained human resources and advances in national industry. Based on Brazilian scientific history and remaining challenges, the key potential areas for future research are (i) the development of digitally based decision support systems, i.e., a shift of focus from on-farm data technologies towards effective, site-specific decision making based on digital data and improved analytics; (ii) on-farm precision experimentation to underpin on-farm data collection and the development of new decision tools; and (iii) novel machine learning approaches to promote the implementation of digitally based decision support systems.

**Keywords:** precision farming; bibliometrics; scientific network; soil health; crop yield

## 1. Introduction

Food demand is expected to double from 2005 to 2050 due to global population growth [1]. Likewise, energy consumption is expected to grow by nearly 50% from 2018 to 2050 [2]. The expansion of agricultural land is the most efficient strategy to increase food and bioenergy production. However, even though agricultural areas are still expected to grow in the coming decades, agricultural expansion to new land (particularly, natural vegetation) is usually associated with negative environmental impacts [3,4]. Therefore, most of such a growing demand should be attended by the sustainable intensification of farming

systems, especially for tropical regions, which are the last frontier for global agriculture [5]. In this scenario, best management practices (e.g., no-tillage, crop rotation, and organic amendments) coupled with suitable technology investments (e.g., precision agriculture (PA)) are imperative to underpin sustainable agricultural intensification [5–8]. PA is a management strategy that takes account of temporal and spatial variability for improved resource use efficiency, productivity, quality, profitability, and sustainability of agricultural production. It comprises a set of technologies that combine sensors, information systems, positioning systems, data processing, and modern machinery to manage spatial–temporal variability [7]. It is a new and ongoing agricultural revolution in information technology that aims to tailor management coherently and holistically, by exploiting the spatial and temporal variability of crop and environmental traits in particular [8]. From a general perspective, PA envisions increasing crop production and also quantifying, monitoring, and managing other essential ecosystem services that support human well-being [9].

PA principles regarding field heterogeneity have been recognized since the 1920s (e.g., Linsley and Bauer [10]). However, field-scale applications started during the 1980s in the U.S. and were adopted in the following years by European countries, China, Australia, India, and South American countries such as Argentina and Brazil [11–15]. The promotion of PA principles worldwide has created new opportunities to take advantage of a combination of newly available technologies, such as global navigation satellite systems, sensors, digital technologies, information, connectivity and communication technology, and the internet of things (IoT) to produce more in a more sustainable way [14,16,17].

Brazil is one of the world's largest producers of agricultural products, presenting remarkable advances in crop yields (increase of 206%) and grain production (increase of 394%) within a cultivated area that has only expanded 61% in the last 40 years [18]. During the 2019/2020 growing season, Brazil reached its highest grain production in history, with 251 million tons produced over 65 Mha (Figure 1). The evolution of Brazilian agriculture has been driven primarily by introducing adapted and improved crop cultivars, conservation soil management, mechanization, public agricultural funding, and investment in technologies such as PA and irrigation [19]. Regarding the country's land ownership structure, we are characterized by a high inequality in land distribution. For example, properties of up to 50 ha represent (in 2017, the last Brazilian Agricultural Census) 81.4% of the total amount of rural establishments and occupy only 12.8% of the country's rural area, while establishments with more than 2500 ha represent 0.3% of the total number and occupy 32.8% of the area [20]. This means that besides the weight of the large agricultural operations in the country, we also have a considerable structure of smallholder farmers.

The introduction of PA techniques in Brazil occurred late in the 1990s by pioneering research conducted at public universities and in the Brazilian Agricultural Research Corporation (Embrapa) [21,22]. Recognizing the spatial variability in crop yield and soil chemical properties have been the two main PA pillars in Brazil. However, over the last 25 years, the science-based knowledge generated in the country has also allowed advances in other areas, such as plant disease management; robotics; proximal, aerial, and orbital sensing; digital platforms; software; and decision support tools [17]. Overall, these advances have followed the global direction towards using more sophisticated and modern tools to optimize financial and environmental assets in agricultural systems [14].

Recently, the status of the scientific research on PA was evaluated by a worldwide bibliometric and social network analysis [23] and by a science mapping approach that emphasized European countries, especially Italy [14]. Moreover, records of completed surveys were examined to document the national- and regional-level adoption patterns of PA worldwide [16] and in Brazil [17]. Despite that, there is a knowledge gap regarding how PA research has evolved in developing countries such as Brazil, one of the global agricultural sectors' leading players. According to this scenario, we conducted a comprehensive and systematic review for synthesizing and discussing the available literature to reveal the trajectory, advances, perspectives, and lessons learned from the past 25 years of PA

in Brazil. Furthermore, the sector organization and national public policies for PA were also reported.

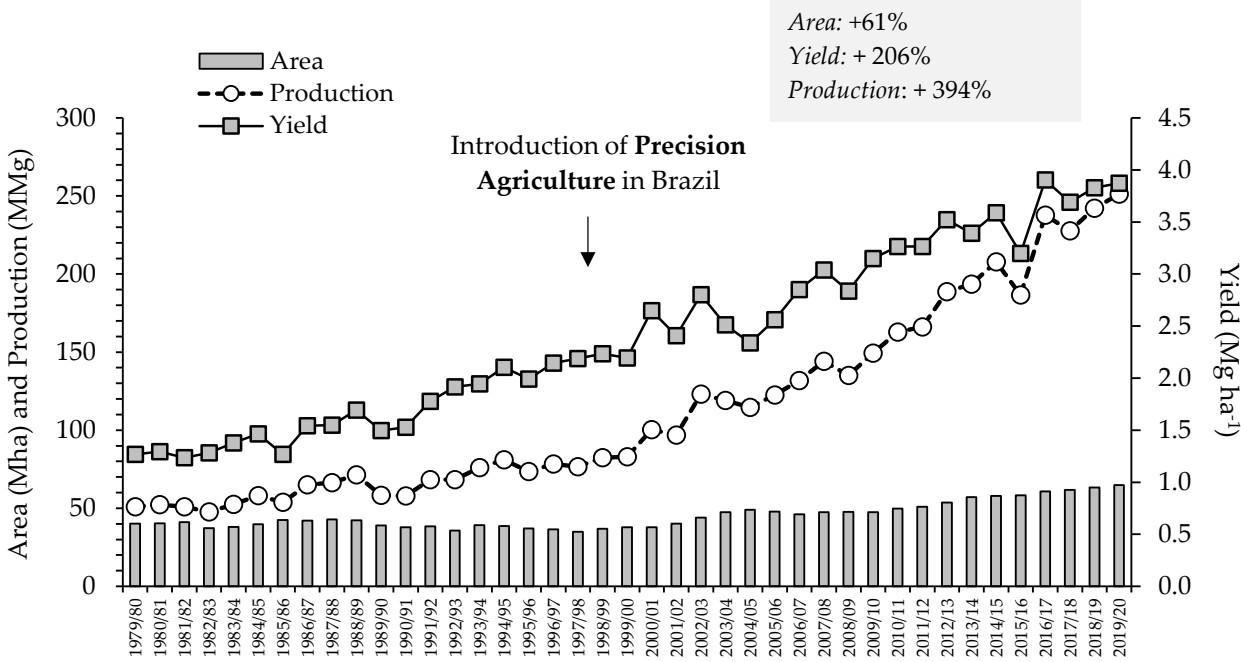

**Figure 1.** Evolution of cultivated area, production, and yield of grain in Brazil in the past four decades. This figure compares the evolution of the cultivated area in Brazil over the last 40 years with its total production and yield, showing the remarkable advances in crop yields (increased 206%) and grain production (increased 394%) within a cultivated area that has only expanded 61%. Adapted from CONAB [18].

## 2. Methodology for Surveying and Analyzing the Scientific Bibliography

The methodology applied in this study is schematically presented in Figure 2. How the systematic review was conducted can be divided into six steps (which are detailed in the sections below): (i) survey of articles in scientific search platforms; (ii) filtering of articles following the pre-established criteria; (iii) compilation of basic information from the papers (e.g., authorship, place of affiliation, year, etc.); (iv) categorization of the papers into the seven areas that comprise the scope of PA; (v) descriptive evaluation of the publications involving spatial and temporal aspects; and (vi) critical evaluation of the literature published in each of the seven areas by considering the trajectory, advances, lessons, and future perspectives. Moreover, at the end of the discussion (Section 3.4), information on the sector's organization and public policies was included to add context to Brazilian scientific production. Finally, in the last section, a general evaluation of the state of the art of PA in Brazil was performed, key areas that need to be further developed were pointed out, and a SWOT analysis of each PA research area was added to summarize the main messages and implications learned from the present study.

### 2.1. Data Compilation

A search of the peer-reviewed literature was conducted to systematically review publications related to PA in Brazil using Web of Science, Scopus, Science Direct, Scientific Electronic Library Online (Scielo), and "Portal de Periódicos CAPES/MEC". The scope of this review was limited to the terms "precision agriculture" OR "precision farming" AND "Brazil", which were searched in the article title, abstract, keywords, and subject within each database. Scielo and "Portal de Periódicos CAPES/MEC" are two databases that include Latin American journals, so the previously mentioned terms were searched in Portuguese.

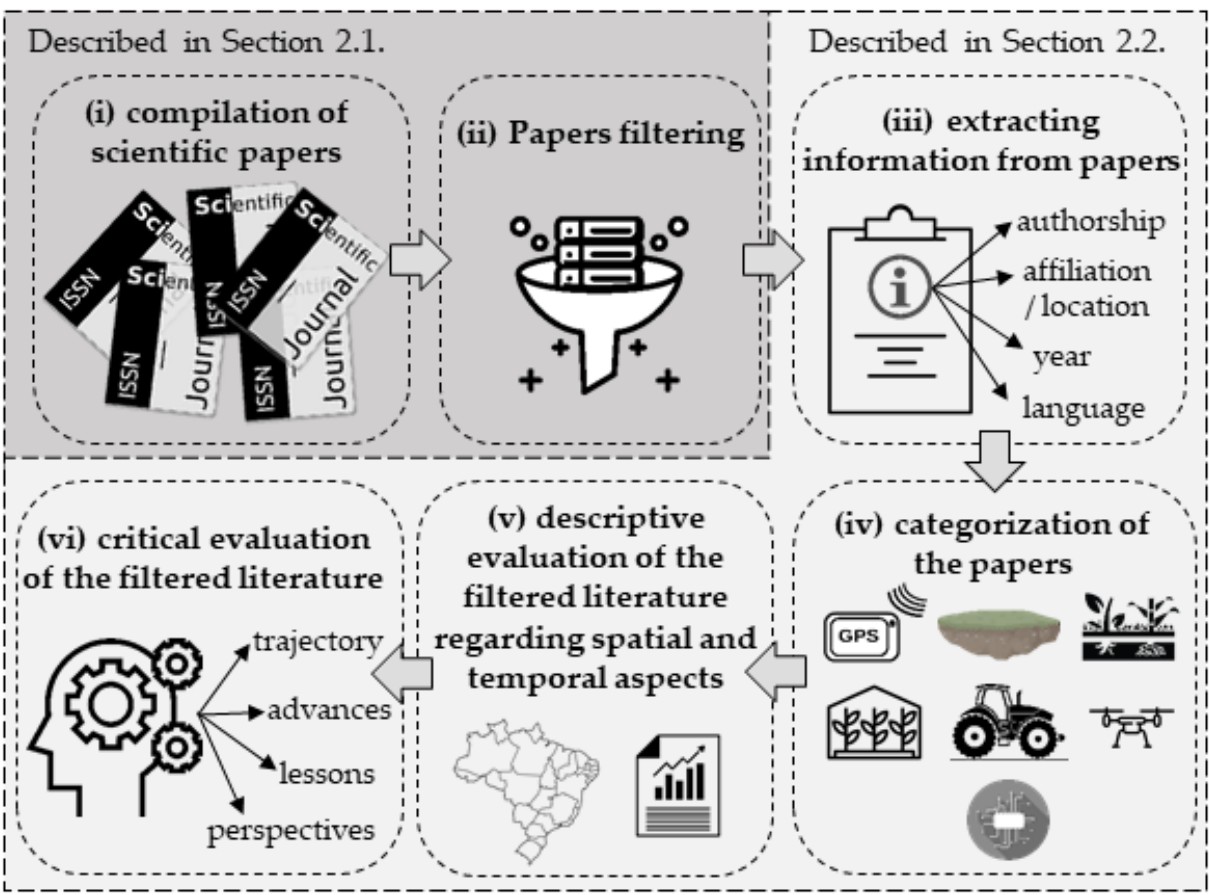

**Figure 2.** Framework of the methodology applied for surveying and analyzing the trajectory of the last 25 years of precision agriculture in the Brazilian scientific bibliography.

The systematic review considered publications published between 1996 and 2020. After the first search, all publications were screened based on the following inclusion criteria:

1.　Peer-reviewed papers excluding grey literature (e.g., conference proceedings, technical reports, books, PhD thesis, and Master's dissertations);
2.　Studies conducted in Brazil involving one or more Brazilian author;
3.　Studies effectively related to PA (concepts, tools, and applications).

Papers that only mention PA in some parts of the text but that did not constitute a study within the PA scope were excluded, as were those related to the area of animal production. In addition, particular attention was paid to avoiding duplicate publications since many articles were found in more than one database. The final database comprised 442 publications (Table S1 in the Supplementary Materials). As mentioned above, books (which are shown in Table S2 in the Supplementary Materials) were not included in the systematic review; however, given their great contribution to the promotion of PA across the country, we considered the main PA books published in Brazil for the discussions on the scientific knowledge generated over the last 25 years.

### 2.2. Data Analysis

The authorship, year of publication, affiliation, and journal information was extracted from each publication to analyze the temporal and geographical/spatial dimensions. The publications were grouped into five periods: 1996–2000, 2001–2005, 2006–2010, 2011–2015, and 2016–2020, and were geographically plotted on a map of Brazil to monitor the regional and national evolution of PA research. The addresses of the affiliations were geocoded using "Google Geocoding API" to obtain geo-referenced locations. Each author's location received a weight that was inversely related to the total number of coauthors in each

publication so that the sum of the weights equaled one. The coordinates and the weights were used to build heatmaps, which were overlaid with the administrative boundaries of countries and states. In addition, the locations of coauthors of the same publication were used to build connections among them. These connections were categorized in national and international collaborations and were represented in maps showing the scientific collaboration network within Brazil and with international researchers.

Based on the title and abstract of each publication, we categorized the publications into seven research areas within the PA scope: (i) global navigation satellite system (GNSS) applications; (ii) soil management (e.g., georeferenced soil testing, soil fertility mapping, and variable rate fertilizer application excluding N); (iii) plant management (e.g., variability of crop yields, crop monitoring through sensors, and sensor-based N fertilizer application); (iv) phytosanitary management; (v) machinery, equipment, and autonomous vehicles; (vi) remote sensing and unmanned aerial vehicles (UAVs); and (vii) decision support tools (i.e., studies focused on data analytics and modeling tools to promote improved on-farm decisions). The number of publications included within each category was recorded, and the relative contributions over time of each area were calculated. Finally, each publication was carefully reviewed to understand and to critically discuss the trajectory of each PA research area in Brazil in the last 25 years. Then, a SWOT analysis (strengths, weaknesses, opportunities, and threats) was performed to synthetize the main messages obtained from this study.

## 3. Results and Discussion

### 3.1. Spatiotemporal Distribution of Precision Agriculture Publications in Brazil

The number of papers published from studies conducted in Brazil significantly increased between 1996 and 2020 (Figure 3); as mentioned above, this review covered a total of 442 papers. Based on the spatiotemporal approach, the distribution of the Brazilian PA publications can be characterized by four phases:

1.  Up to 2000: In the first years after the introduction of PA in Brazil, only a few studies were published (totaling nine publications), mainly by pioneer PA research groups explicitly located in the southeast (i.e., São Paulo state) and south (i.e., Rio Grande do Sul state) regions of Brazil (Figure 4). In 1995 and 1999, the first PA symposia were organized at ESALQ/USP. In addition to the initiatives led by public universities, in this period, Embrapa also started to develop multiple projects related to PA into the National Agricultural Automation Program (called Program 12) [24] and the Brazilian Agricultural Technology Development Support Project—Prodetab, as reported by Inamasu and Bernardi [25]. In 1999, Embrapa published a document listing the infrastructure of PA in Brazil, including active researchers, companies, equipment, software, publications, and webpages related to PA [26].

2.  2001–2005: A slight increase in the number of PA papers occurred in the early 2000s (Figure 4), with 33 papers being published in this period. This was boosted by the diffusion of PA concepts through classic textbooks and book chapters (e.g., Balastreire [27]; Borem et al. [28]; and Molin [11]), the disabling of GPS-selective availability signals making it more accessible and cheaper, the development of the first national equipment for the variable-rate application of fertilizers, and the field-application of PA by the first service providers. Furthermore, several initiatives emerged during this period that were led by universities, Embrapa, and private companies, such as Projeto Aquarius (https://projetoaquarius.agr.br/, accessed on 31 October 2022), which was created in 2001 by the Federal University of Santa Maria (in southern Brazil), and the 5-year PA macroprogram (phase 1), which was created by Embrapa in 2004. The Federal University of Viçosa (UFV) organized the International Symposium on Precision Farming (SIAP) in the years 2002, 2005, and 2007. The first Brazilian conference on PA was organized in 2004, a biannual event that included both the scientific community and industry in the same environment. Therefore, in the early 2000s, existing research groups had been consolidated, and new groups had emerged in the south, southeast, and central-west regions of Brazil (Figure 4).

3.  2006–2010: The number of PA papers increased more significantly between 2005 and 2010 (Figure 3), with 80 papers being published in this period alone. Studies on PA were intensified in this period, driven by the diffusion of PA among the scientific community, crop consultants, and farmers, which is a sign of the solidification of the structure built in previous years. The 5-year PA macroprogram (phase 2) of the Embrapa was also launched in 2009, gathering dozens of researchers to work in diverse areas of PA. In terms of the spatial distribution of publications, it was observed that a small proportion of these publications started to come from the northeast region of Brazil (Figure 4).

4.  2011 to present: The last decade has been marked by the most significant increase and internationalization of Brazilian PA publications (Figure 3). This increase in internationalization is likely due to the "Ciência sem fronteiras" government program ("Science without borders"; [29]) between 2011 and 2020, which funded undergraduate and PhD Brazilian students to carry out part of their research education at foreign institutions. The recent expansion of Brazilian science-based knowledge in PA, with 155 papers being published in the first quinquennium and 165 being published in the second, has been driven by multi-dimensional factors such as (i) investments in education that have promoted the expansion of public universities to the interior of the country (e.g., Support Program for Restructuring and Expansion Plans of Federal Universities—REUNI [30]) and the creation of new PA research groups (Figure 4); (ii) the implementation of PA as a subject integrated into agronomy and related under- and graduate courses [31] coupled with the publication of updated textbooks on PA (e.g., books mentioned in Table S2) with consolidated concepts and applications of PA for students and the broader community; (iii) public funding for research (for example, the Embrapa's PA macro programs [25]); (iv) incentives for international scientific collaboration; v) the creation of PA-related commissions and associations (e.g., Brazilian Commission of PA (CBAP) into the Brazilian Ministry of Agriculture, Livestock and Supply (Brasil, 2012), the Brazilian Association of PA (AsBraAP—https://asbraap.org/, accessed on 31 October 2022), and the Brazilian Association of Service Providers in PA (ABPSAP—https://www.abpsap.org.br/, accessed on 31 October 2022) as well as National Strategic Agenda for PA—2014–2030 [32]; and (vi) the development and popularization of new technologies applied to agriculture (e.g., Internet, smartphones, drones, machine learning, etc.).

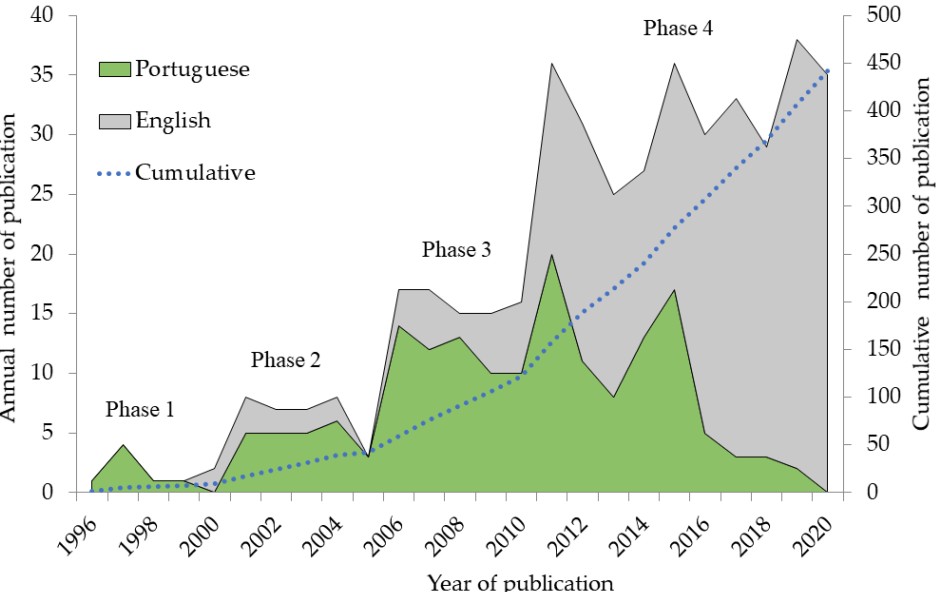

**Figure 3.** Number of publications in precision agriculture over the past 25 years in Brazil. Four phases of PA research in Brazil were identified based on the evolution of the number of published papers.

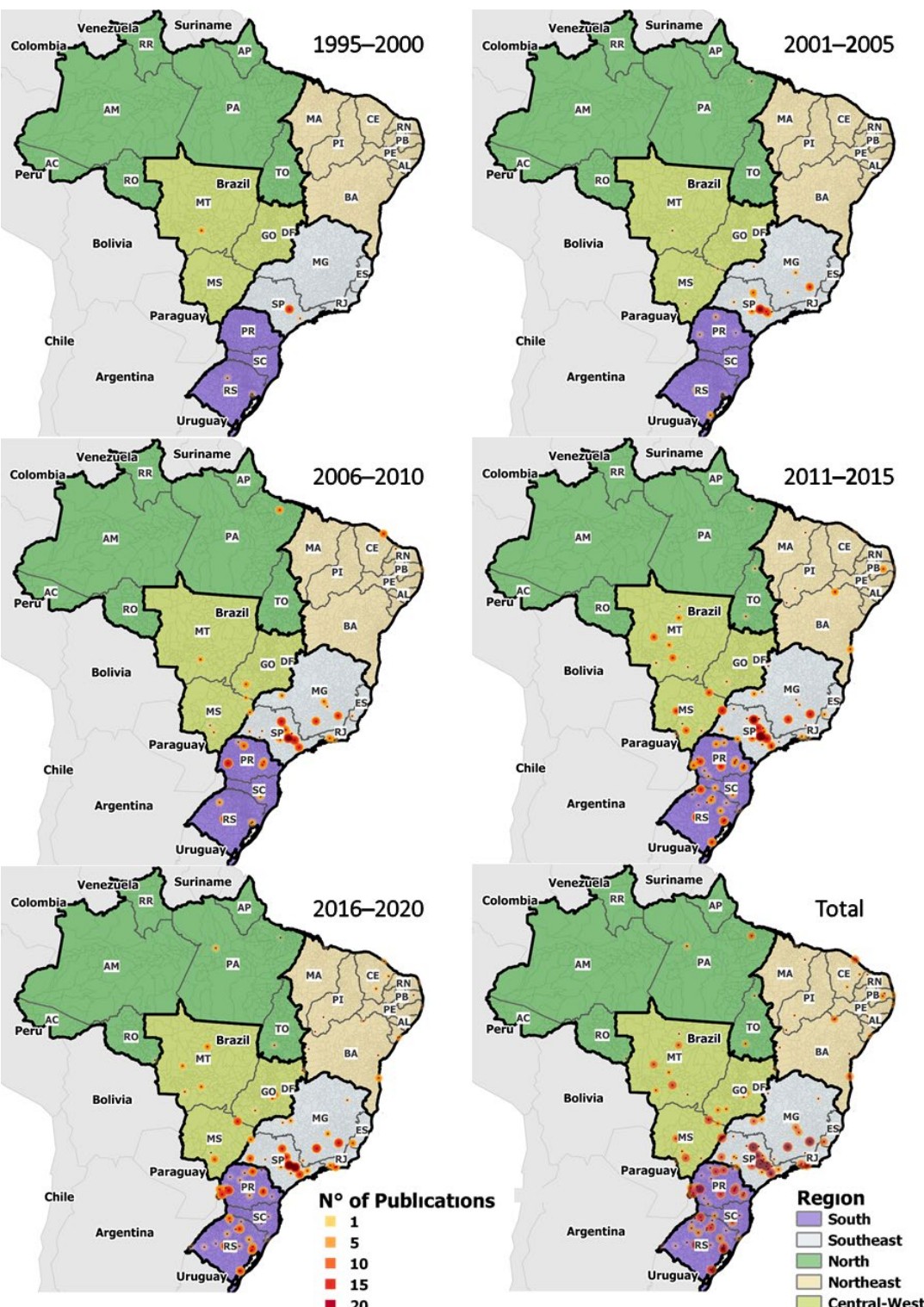

**Figure 4.** Spatiotemporal distribution of precision agriculture publications over the past 25 years in Brazil. The color and size of the circles are proportional to the frequency of publications in each location. The Brazilian states of Acre (AC), Amapá (AP), Amazonas (AM), Pará (PA), Alagoas (AL), Bahia (BA), Ceará (CE), Maranhão (MA), Paraíba (PB), Pernambuco (PE), Piauí (PI), Rio Grande do Norte (RN), Sergipe (SE), Distrito Federal (DF), Goiás (GO), Mato Grosso (MT), Mato Grosso do Sul (MS), Espírito Santo (ES), Minas Gerais (MG), Rio de Janeiro (RJ), São Paulo (SP), Paraná (PA), Rio Grande do Sul (RS), and Santa Catarina (SC) are indicated with their acronyms.

It is important to mention that there was also an increase in academic reports in the area of agriculture as a whole (Figure A1), which leads us to think that this increase in publications in the area of PA may be related to a natural rise in the number of publications. However, it is worth noting that in 1996, PA was a new area in Brazil, and no publications related to this topic existed before that date. When comparing the evolution of publications in percentage, for the areas of agriculture and PA, according to the number of publications on each topic in 1996 (Figure A1C), we can observe that from the year 2000, the rate of increase in PA publications is slightly higher than that observed for agriculture. This may represent an increase in interest in investigating PA-related topics in a way that is disconnected from the natural increase that has occurred in recent decades. Another practical aspect that represents this specific increase in interest is the growing number of people who have participated in the Brazilian conference on PA (ConBAP) since its creation in 2004 (with 291 participants in 2004 and 664 in 2022) as well as the rising number of associates joining AsBraAP (with 11 associates in 2016 and 158 in 2022).

*3.2. Scientific Collaboration Network on Precision Agriculture*

National scientific network related to PA is predominantly concentrated in the southeast and south regions of the country (Figure 5), where the more traditional PA research groups are located. The southeast region (especially in São Paulo State) is the principal hub connecting PA researchers from regions all around the country. There is an evident difficulty in the interactions among researchers located in areas with a lower number of publications (i.e., central-west, northeast, and north), suggesting a strong linkage or even dependence from researchers located in southern regions.

Our findings showed that 36% of the studies evaluated ($n$ = 161) included international collaborations (Figure 5). The largest number of collaborations was established with researchers from the U.S. and European countries, especially Spain, Germany, and the UK. These results are aligned with those reported by Aleixandre-Tudó et al. [23], who showed that these countries occupy a central position in the global scientific collaboration on PA. Few interactions were observed with researchers from other continents, highlighting initiatives with China, Colombia, and Australia.

This overview of scientific PA connections in Brazil can guide the National Agenda of PA and its multiple actors to prioritize efforts to promote PA principles in strategic regions that are still uncovered. For example, Brazilian agriculture frontiers are located along the Amazon/Cerrado borders (e.g., Rondônia and Pará States) and more recently in the northeast region, including parts of the states of Maranhão, Tocantins, Piauí, and Bahia (called the MATOPIBA region) [33]. However, the limited PA research initiatives in those regions (Figures 3 and 4) likely represent a barrier to adopting locally adapted technologies, which could contribute to more efficient and sustainable agriculture expansion [6,8,34]. Our findings can also be useful to elaborate strategic planning to promote and prioritize interactions among PA researchers and institutions from a local to a global scale, as recently stressed by Aleixandre-Tudó et al. [23] and Pallottino et al. [14].

*3.3. Advances in and Perspectives on Precision Agriculture Research Areas in Brazil*

The temporal evolution of scientific publications in each PA research area is shown in Figure 6. Out of the eight papers published until 2000, three were conceptual reviews (i.e., Molin [21,22,35]) that aimed to present PA principles and potential applications for the Brazilian audience. At that early stage, field research focused on soil management, soil testing, the adaptation and development of machines and equipment, and applying global positioning systems—GPS (Figure 6). In the early 2000s, the research advanced to new areas, such as plant management, phytosanitary applications, sensors, robotic and remote sensing, and the use of remotely piloted aircrafts (UAVs).

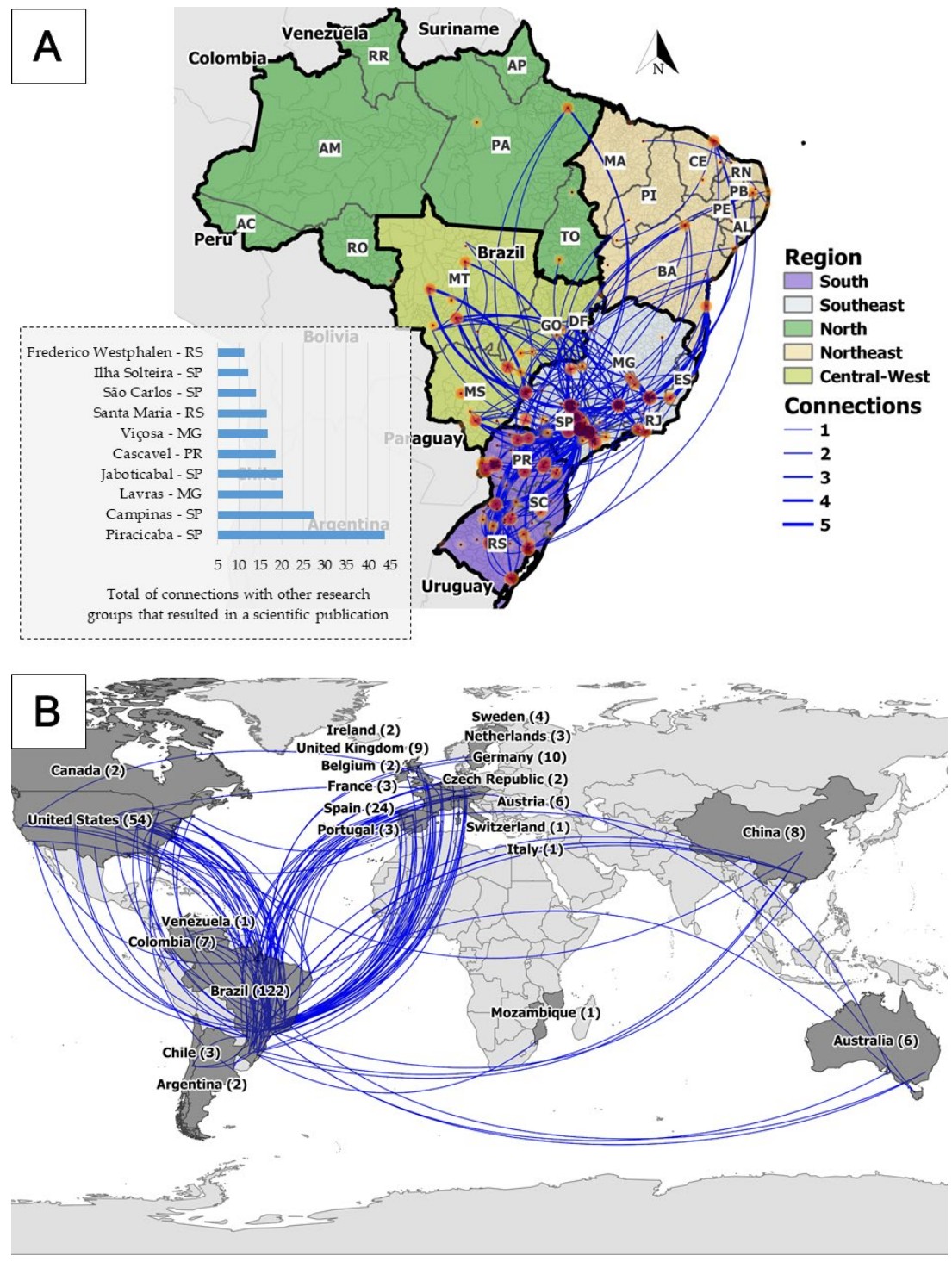

**Figure 5.** Brazilian scientific collaboration network associated with the precision agriculture research field showing national (**A**) and international (**B**) interactions; a bar graph showing the total connections of the top 10 Brazilian cities holding the highest number of connections is also presented (this information is also shown in detail in Figure A2). Each blue line represents a paper co-authored by researchers from two different cities/countries. The total number of connections for each country has been indicated in brackets. The Brazilian states of Acre (AC), Amapá (AP), Amazonas (AM), Pará (PA), Alagoas (AL), Bahia (BA), Ceará (CE), Maranhão (MA), Paraíba (PB), Pernambuco (PE), Piauí (PI), Rio Grande do Norte (RN), Sergipe (SE), Distrito Federal (DF), Goiás (GO), Mato Grosso (MT), Mato Grosso do Sul (MS), Espírito Santo (ES), Minas Gerais (MG), Rio de Janeiro (RJ), São Paulo (SP), Paraná (PA), Rio Grande do Sul (RS), and Santa Catarina (SC) are indicated with their acronyms.

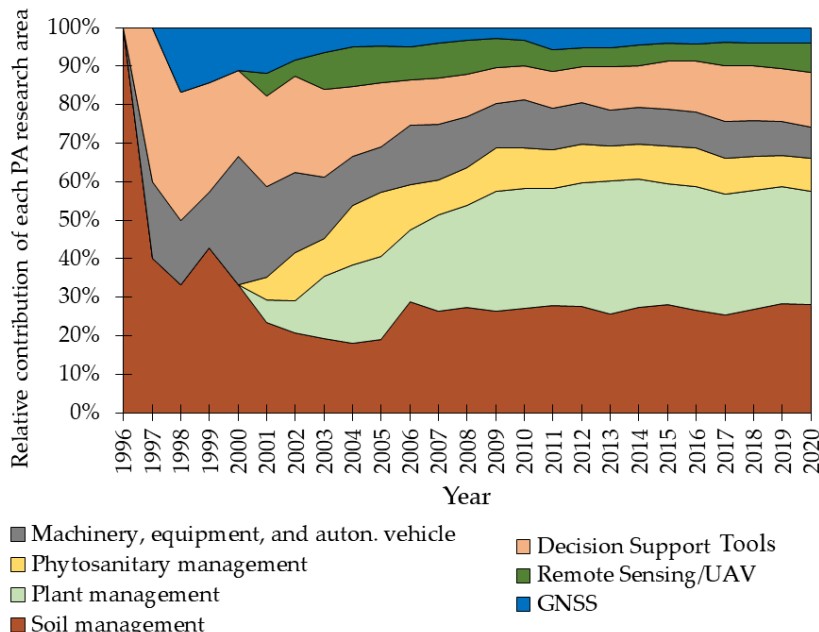

**Figure 6.** Temporal evolution of the number of publications in each PA research area in Brazil.

Overall, soil and plant management remain the predominant areas of PA research, accounting for 57.5% of the total publications published until 2020 (Figure 7). If phytosanitary management was included in plant management, then the total contribution of the soil and plant areas would increase to 66.1%. Publications on decision support tools have also grown in recent years, totaling 14.3% of the total number of studies (Figure 7), which could be related to the advent of research and commercial solutions in form of connectivity, mobile apps, digital platforms, and software.

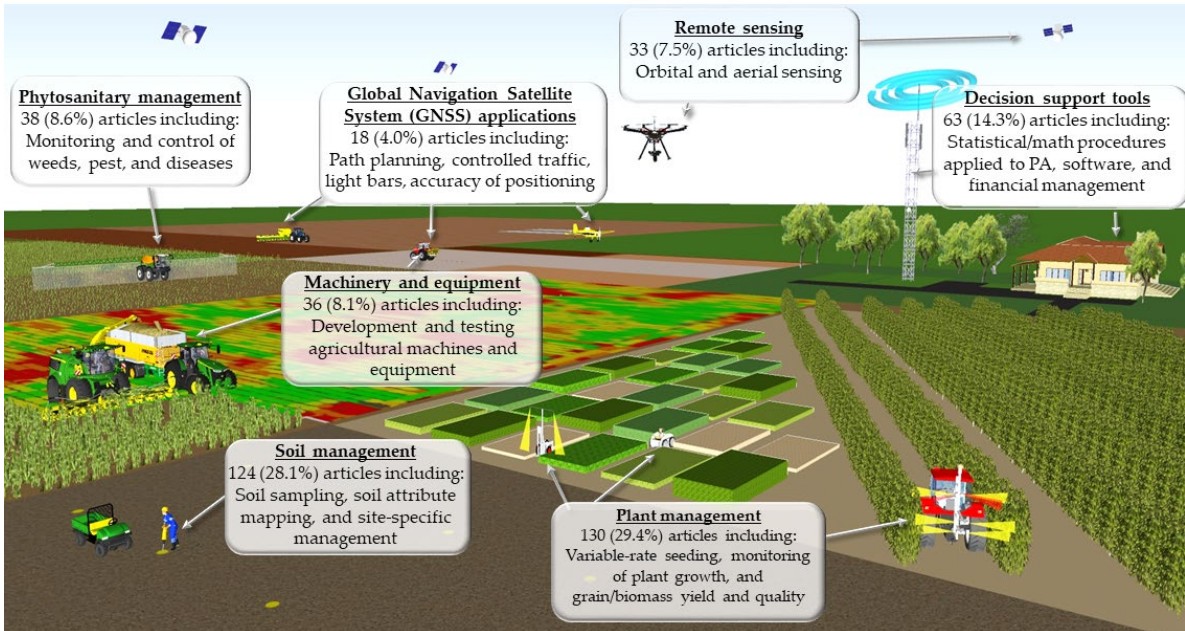

**Figure 7.** Precision agriculture research areas and their absolute and relative contributions to the total of publications included in this study.

In the following subsection, we describe and discuss how the technologies have evolved in the past 25 years of PA experience in Brazil. The trajectory, advances, lessons learned, and future perspectives within each PA research area are pointed out.

### 3.3.1. Global Navigation Satellite System (GNSS) Applications

The contribution of the Global Navigation Satellite System (GNSS) in the emergence and adoption of PA principles is undeniable [16]. Initially, this contribution comes from the ease of geo-referencing field data. Likewise, more impactful applications have been seen, such as machine orientations for performing localized operations (e.g., sowing, fertilizing, spraying, and harvesting). Location technologies have evolved, and agriculture has continued to demand cinematic positioning with greater accuracy, requiring practical and high-accuracy positioning systems. At the end of the 1990s, Molin [35] indicated the need to implement solutions for differential correction and to know the quality of the positioning of GNSS receivers for agricultural applications. Some authors proposed an instrumented vehicle to obtain the performance of the moving receivers [36] since the accuracy dilution caused by this cinematic characteristic was not considered by the national market when evaluating receiver performance.

The light bar was the first solution to use GNSS as a guide and was initially used for agricultural aircrafts according to its straight, parallel, and equidistant strides. In the early 2000s, self-propelled sprayers were the first agricultural vehicles to incorporate this technology. Baio et al. [37] concluded that the light bar was more efficient and practical than foam markers to guide pesticide applications. Automatic steering systems arrived in Brazil at the turn of the century. This technology associated with GPS-RTK was evaluated by Baio and Moratelli [38] for the mechanized planting of sugarcane, and it was observed that the accuracy of parallelism between steps was five times better than that obtained with manual targeting.

The ionosphere over Brazil is heavily affected by intense scintillation conditions [39]. It is a limitation that can lead to communication signal losses by receivers, resulting, e.g., in widespread interruptions of mechanized operations during periods in which this phenomenon has increased intensity. Mitigation strategies can reduce part of this problem [40]; however, unfavorable scintillation conditions should be monitored, and operations should be avoided under these conditions. On the other hand, in recent years, multi-constellation receivers have become more available, which has enhanced redundancy and accuracy in environments with limited signal visibility [41]. In addition, it is expected that the combined use of different GNSS (e.g., GPS + Beidou/Compass + Galileo Systems) associated with the modernization of communication signals will increase the overall performance and robustness of satellite navigation for PA applications, avoiding positioning errors that, although accepted for certain operations, could be reduced.

### 3.3.2. Soil Management

Brazilian tropical agriculture is mostly developed on highly weathered, low-fertility, and acidic soils. This explains the high demand for these inputs by the country—the world's largest consumer of fertilizers in 2021 and 2022—and why the variable rate application of fertilizer and lime is the most widely adopted PA practice. Not coincidentally, research involving soil management has been broadly covered since the beginning of PA in Brazil (Figure 6). The first scientific paper published in Brazil on PA evaluated the spatial relationship between soil properties and corn yield [42].

Site-specific soil management in Brazil is strongly reliant on soil mapping using systematic georeferenced soil sampling (grid sampling) followed by the interpolation of the results obtained from soil laboratory analysis. Since the early years of PA, several studies have investigated the characteristics of spatial variability of the chemical [43–46], physical [43,47], and, more recently, biological [48,49] attributes in Brazilian soils. Still, in the 2010s, some authors sought to define an "optimum" grid size (i.e., sampling density) for mapping the spatial variability of soil fertility attributes using geostatistical methods, in view of the expansion of this approach on Brazilian farms [50,51]. These studies agreed that although spatial variability varies according to inherent soil properties and management practices, sampling grids larger than 100 m $\times$ 100 m (1 sample ha$^{-1}$) are often inefficient for characterizing most of soil fertility attributes in Brazilian agricultural fields [50]. Furthermore, attributes that exhibit more abrupt spatial variations, such as K and P (e.g., between <10 and 68 m [51,52]) and penetration resistance (e.g., <30 m [53,54]), require sampling at an even higher spatial density for reliable mapping. Despite this scientific

consensus, it is common for PA users (e.g., farmers and consultants) to create maps with low spatial density sampling, usually smaller than <0.5 sample ha$^{-1}$ (i.e., sampling grids larger than 140 m × 140 m) [51]. Therefore, there is still an urgent need for Brazilian research to practicably demonstrate the limitations of such approaches and the benefits of alternative site-specific soil management practices.

Soil sensing technologies are promising tools to improve the characterization of the spatial variability of soil attributes [55,56]. Starting in the 2000s, many studies have been conducted in Brazilian soils to evaluate proximal and remote sensing techniques to characterize agricultural fields with a high spatial density of information [57,58]. Studies evaluating proximal soil sensing techniques have mainly covered apparent electrical conductivity (ECa) sensors [59], with few studies evaluating on-the-go applications of ion-selective electrodes (ISE) [60] and the diffuse reflectance spectrometers that work in the visible and near infrared (Vis-NIR) regions [61]. Most studies have evaluated the predictive performance of these sensors for determining soil attributes of agronomic interest. Some authors have even proposed variable-rate application maps based on sensor outputs [61], but no work shows the long-term effects of sensor-based localized soil management. On the other hand, novel applications of ECa data have been presented for tropical fields. Sanches et al. [62] used ECa data to drive targeted soil sampling and then used both the sensor output and the soil laboratory analysis to create maps using kriging with external drift. This approach allowed for more assertive mappings with a smaller number of samples when compared to high-density sampling performed using regular grids. Sanches et al. [63] demonstrated the potential of ECa data to assist in the classification of production environments for sugarcane cultivation. Additionally, with the potential to aid in the interpretation of soil variability, some studies have explored topographic parameters for designing targeted soil sampling [64] since higher soil spatial variability is associated with convex and concave relief when compared to linear relief shapes [65,66].

Contributions have been made by Brazilian researchers to develop quick and reagent-free soil characterization procedures using sensing techniques in laboratory environments, with emphasis on diffuse reflectance spectroscopy in the Vis-NIR and mid-infrared (MIR) spectral regions [57,67–69] and using energy-dispersive X-ray fluorescence (ED-XRF) spectroscopy [70,71]. In this context, Demattê et al. [72] proposed the concept of hybrid laboratories, which represents a clear evolution of traditional soil testing by integrating sensing techniques and can provide laboratory analyses with better quality (e.g., identifying outliers), greater agility, and more affordable prices. The development of regional spectral libraries for the implementation of hybrid laboratories and in situ analysis should drive some research fronts in the coming years.

Although there is consensus in the Brazilian literature that effective soil fertility characterizations require strategies for data collection at fine resolutions, few scientific works have used modern sensing and data analysis technologies to promote advances towards a more efficient characterization of soil variability. Technologies related to multi-sensor platforms (e.g., Guerrero et al. [73]) and multivariate geostatistical approaches (e.g., Castrignanò et al. [56]) are underexplored in the Brazilian context. Future research should create and evaluate strategies to integrate fine-scale soil data with multi-causal decision-making systems to propose novel approaches for site-specific management in tropical soils.

### 3.3.3. Plant Management

Recognizing and mapping the spatial variability of crop yields were the first targets of plant-related PA research in Brazil. In this context, Balastreire et al. [74] published the first yield map in a corn field in central Brazil. Despite the utility of recognizing spatial variability of crop yield, the massive adoption of yield mapping is not a reality in Brazilian farms, contrary to what happened in Argentina (e.g., Lowenberg-Deboer and Erickson [16]). The limited adoption of yield mapping can be associated with three main reasons: (i) the high cost of the technology, although it is becoming less expensive [17]; (ii) the limited number of trained professionals to collect and store the data properly; and (iii) the sense of a lack of utility, since most of the users only see the yield map, but they do not use it to support decision making with regard to

land use and management practices. In many cases, data are collected by farmers but turn out to be useless due to inconsistencies in the datasets (e.g., positioning errors, incorrect platform width, data overlapping, outliers, absence of data, calibration inconsistencies, etc.), as previously reported by Menegatti and Molin [75]. An overview of the main difficulties and challenges to strengthen the use of crop yield mapping and other PA technologies by Brazilian farmers can be consulted in Bolfe et al. [17].

Several studies have been performed to identify the soil-limiting factors associated with crop yield variations. Santos et al. [76] conducted a pioneering study to correlate corn yield changes with the chemical and physical attributes of soil. Subsequently, similar studies were conducted for other crops, such as coffee [77], black beans [78], fruit species [79], sugarcane [80], soybeans [81], and pasture [82]. Similarly, the spatial variability of crop yields also started to be investigated under additional parameters of sowing performance, such as seeding depth [83] and spacing [84], plant population [85,86], and yield components [84]. These studies resulted in insights into how crop yield variability could be explained by soil and climate parameters and into the quality of sowing operations and stand of plants. PA research in fruit crops is still less developed than in annual crops. Nonetheless, significant advances have been achieved by mapping yield; the leaf-tissue nutrient concentration; and the number, weight, and quality of fruits in apples and orange orchards [87–90].

The first research using optical canopy sensors for crop variability assessment with high-density data collection was published in the early 2010s [91]. In general, the first studies are characterized by attempts to adapt the sensors and methodologies developed in other countries (i.e., the U.S. and European countries). Crop sensing technologies to aid in-season nitrogen prescription in wheat [92], sugarcane [93], and cotton [94] had been tested in Brazilian fields. The authors evaluated different commercial sensors and calibration strategies varying from simple redistributions to previously calibrated agronomic algorithms, which, in most cases, is associated with the use of "N-rich strips". Vegetation indices were applied to create management zones for site-specific fertilization interventions (e.g., Amaral et al. [95]) and even to design crop rotation plans [96]. Sonar and LiDAR (Light Detection and Ranging) have also been used as tools for extracting biometric variables, such as canopy height and volume from orange orchards [90] and plant height for the localized application of plant growth regulators in cotton [94] (Figure 8).

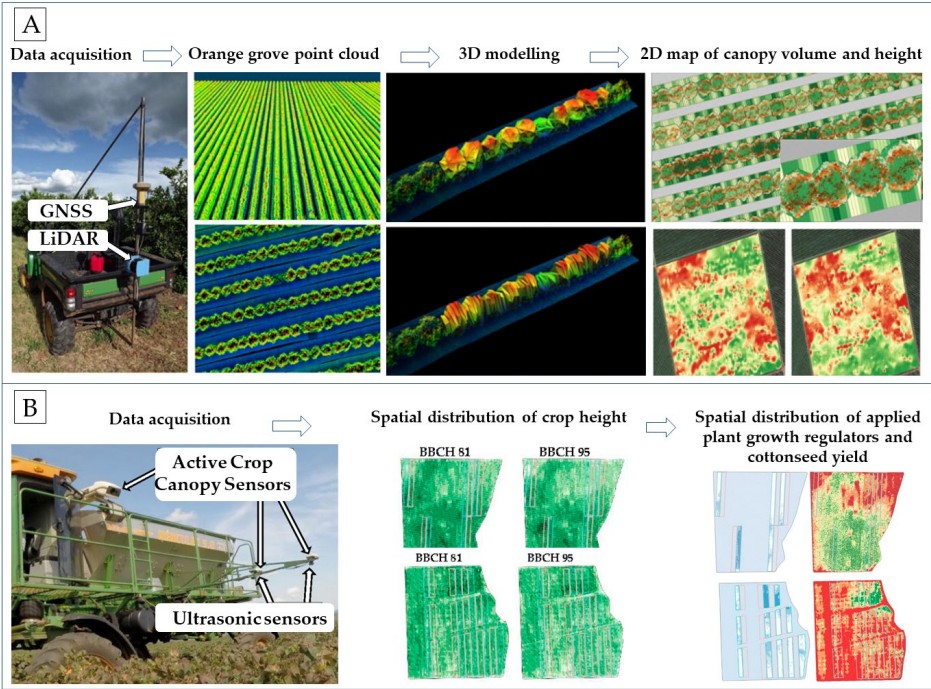

**Figure 8.** Advances in modeling crop canopy in Brazil: (**A**) Orange crop geometry information using a mobile terrestrial laser scanner and 3D modeling (adapted from Colaço et al. [90]); (**B**) management of plant growth regulators in cotton using active crop canopy sensors (adapted from Trevisan et al. [94]).

### 3.3.4. Phytosanitary Management

Within the crop management area, phytosanitary research has stood out in Brazil since the beginning (early 2000s) via the geo-referencing and mapping infestation of weeds, pests, and diseases. Early studies showed the temporal stability of the occurrence of weeds, allowing mapping and intervention within a reasonable time interval. However, the monitoring and control of pests and diseases that are influenced by population dynamics and climatic conditions are still challenging.

Balastreire and Baio [97] were pioneers in proposing a rapid methodology using a quadricycle with GNSS to map weeds. Subsequently, several studies focused on mapping and controlling weeds in different cultivation systems, suggesting herbicide savings of 18 to 44% in soybean crops [98]. Although it showed significant results in terms of herbicide savings, manual mapping is a time-consuming and tedious task that limits its application on a large scale. In addition, subsequent research showed that the spatial distribution of weeds was highly dependent on the sample density employed, the cultivation system, and the plant species studied [99,100].

Concomitantly, studies started using remote sensing tools as promising alternatives to the traditional manual sampling of weeds. Viliotti et al. [101] developed a low-cost system capable of scanning and discriminating plants based on the electrical resistance obtained in a Light Dependent Resistance (LDR) sensor coupled with solenoid valves to activate site-specific spraying. Using the principles of the proximal sensing of plants, Merotto et al. [102] evaluated vegetation indices obtained by commercial sensors as a strategy for determining the location of weeds in the field. Likewise, Silva Jr. et al. [103] used a computer vision system to analyze images and to differentiate weed species from dry bean plants. The first study related to pest mapping and control in Brazil was carried out by Farias et al. [104], who analyzed the spatial distribution of soil nematode infestation in cotton. More recently, Martins and Galo [105] used remote sensing to identify the spectral behavior of sugarcane plants attacked by the nematodes and larvae of the *Migdolus fryanus* beetle, discriminating healthy plants from those attacked by means of the carotenoid index (ratio between reflectance at 800 and 470 nm). In fact, soil pests have slow dispersion, presenting greater mapping potential with the aid of remote sensing or UAV images. However, studies focused on aboveground pests also have raised the interest of researchers. Sena Jr. Et al. [106] and Farias et al. [107] sought to identify the damage caused by armyworm (*Spodoptera frugiperda*) in corn plants using digital images and by spatial mapping, respectively. Riffel et al. [108] and Aita et al. [109] studied the spatial distribution of caterpillars (*Anticarsia gemmatalis* and *Pseudoplusia includens*) in soybean fields. These studies showed the aggregated pattern of distribution of these pests in the field, making mapping and localized control an important ally for integrated pest management and to reduce the use of insecticides. Pests such as *Sphenophorus levis* in sugarcane [110] and termites (*Syntermes spp.*) in eucalyptus plantations [111] also had their spatial distribution studied, showing promising potential for reducing costs with chemical control through localized spraying. Finally, spectral information from visible and infrared regions was also applied to detect and map symptoms caused by nematodes in coffee plants [112] and by the mosaic virus in sugarcane [113]. Nevertheless, spectral patterns from visible and infrared regions are strongly influenced by changes in plant biomass and, typically, are only able to detect the late effects of damage caused by pests and diseases. Therefore, recent studies suggest the use of thermal cameras embedded in aerial platforms to assess the surface temperature of plant canopies as an alternative for monitoring diseases in early stages [114] and for promoting the more rational use of pesticides.

Detection and mapping the occurrence of diseases is one of the most complex tasks due to host–pathogen–environment interactions. Yet, only a few publications on this topic were found. For instance, Belasque et al. [115] studied the potential of using fluorescence spectroscopy to detect citrus canker, a disease caused by the bacterium *Xanthomonas citri*, in the early stages. Since the control of citrus canker is carried out by an official eradication campaign, detecting the disease at early stages is crucial to prevent its spread in the field.

More recently, Boechat et al. [116] determined the incidence and severity of white mold (Sclerotinia sclerotiorum) in black beans using multi and hyperspectral sensing. The authors concluded that the results are more promising when using vegetation indices that are not influenced by the soil surface. Based on our results, it was clear that the development of new technologies that improve crop disease control must be promoted in Brazil, aiming to reduce the use of pesticides and its associated economic and environmental costs.

### 3.3.5. Machinery, Equipment, and Autonomous Vehicles

The agricultural machinery industry has been one of the most significant drivers of PA adoption, aiming to manage spatial variability of crop production and to optimize operational efficiency. Agricultural machines have a high level of embedded technology (e.g., different sensor and actuator systems), providing valuable high-density information for site-specific management [117].

The introduction of yield monitors in harvesters was a milestone for establishing a knowledge base for the spatial variability of Brazilian crops. Globally, yield monitors appeared on the market in the early 1990s, initially on grain harvesters. In Brazil, a significant contribution was made by Balastreire et al. [74], who built a system for the continuous weighing of the harvested product composed of a sub-tank inserted inside the grain tank of a harvester and supported on load cells. Yield mapping also provided advances in the management of mechanized harvesting operations.

Yield mapping for sugarcane happened much later than in grain crops and only became mainstream recently [117]. Without adequate tools to measure the spatial variability of sugarcane yield, site-specific technology adoption for this crop has developed more slowly. On the other hand, the cost associated with mechanized operations accounts for most sugarcane production costs, which prompted sugar mills to be early adopters of innovations to increase operational efficiency. This includes most of the high-precision GNSS systems (RTK and subscription-based satellite correction), assisted driving technologies, fleet management systems, and telematics [118]. Sugarcane production has also been at the forefront of controlled traffic farming (CTF) adoption in Brazil, with proven benefits of reduced ratoon crop damage, improved soil quality, and increased longevity (number of harvested crop seasons before replanting) [119]. The path misalignment errors between the tractor and sugarcane trailers are strongly related to the terrain's lateral slope and the path type. In general, it is essential to use RTK guidance in the harvester and the tractor pulling the trailers. In curved tracks in terrain with lateral slopes, it is also necessary to use active auxiliary solutions to tractor automatic steering to avoid lateral shifts in the machinery's set directions [120]. Even though CTF results are encouraging in sugarcane fields, its adoption by grain farmers has not been expansive. The main challenges are related to difficulties in standardizing machinery widths, the common use of old and new equipment on the same farm, and the use of third-party service providers for harvesting.

The use of variable rate applications intensively increased during the first decade of the 2000s, when several controller solutions came to the market. Initially sold as kits to be installed on existing equipment, they became part of the machines directly from the industry. One useful piece of information from the majority of those controllers is the as-applied map [121]. Not sufficiently known yet by users, it represents the operation's spatial report and can help farmers or managers to continuously improve the quality of input applications.

Worldwide, significant advances toward machine automation began in the 1990s, when PA became a key driver for developing and applying intelligent machines (including robotic devices). Despite that, Brazilian research in this area focused on evaluating the benefits of automation (e.g., variable rate application, boom section control) and its influences on crop production. In the 2010s, studies were carried out to assess sprayed volume savings when using automatic section control in sprayers [122] and to determine the effects on corn yield promoted by the control of sowing overlap and the optimization of within-row plant spacing accomplished by precision planters equipped with automatic section

control and vacuum meter systems [123,124]. Baio et al. [125] used a low-cost electronic controller for flow rate (i.e., instead of using direct injection systems, which are traditionally more expensive) to evaluate the management effects of the variable rate applications of phytosanitary products on cotton. The authors concluded that this alternative allowed satisfactory results to be reached if the rate variation did not exceed 20% of the average application rate to maintain the droplet size and spray quality. Similar systems have been used to manage plant growth regulators and defoliants in cotton, fungicides in corn, and desiccants in soybean [94,125].

Scientific publications do not tell the full story of PA machinery development since many national companies, which have internal research and development teams, usually develop patented solutions that are not publicly disclosed in peer-reviewed articles. The largest market share of national equipment includes sprayers, spreaders, and no-till planters. Many national variable rate controllers have been developed, and global companies have acquired other companies. Many products are a combination of national technology, usually for the mechanical parts, with imported electronics such as the GNSS system, controllers, and seed distribution systems. The most common equipment on the market addresses the needs for the variable rate application of lime, phosphorus, potassium, nitrogen, and seeds. Section control for sprayers, spreaders, and planters is also common. More recently, nozzle control and pulse-width modulation (PWM) valves for sprayers have been introduced on the market.

Innovation in the national agricultural machinery industry also includes solutions for the improved quality of variable rate applications at a large scale, with gravity-flow spreaders that are up to 12 m wide for lime application and pneumatic systems for granular fertilizer application of up to 30 m. Another example is a hybrid sprayer/spreader, which fills the gap of using self-propelled high-clearance vehicles in smaller farms that cannot afford to buy multiple machines and use each one only for short periods during the season.

Small and intelligent machines (robots) can be used for many of the PA practices adopted in the Brazilian context, such as soil sampling/analysis, crop scouting, site-specific weed control, among others. Robotic applications might be also more economically feasible than conventional machines for some agriculture applications [126]. However, only a handful of the Brazilian studies that have been published in scientific journals (e.g., Tabile et al. [127] and Barbosa et al. [128]) involve intelligent robotic devices and agricultural machines. One of the critical areas of interest for agriculture robots is integrated pest and disease management programs to help with field scouting, as discussed above. There is also demand for autonomous vehicles at breeding trials and seed production sites, which employ a large amount of manual labor that is in shortage in some places. In this aspect, most of the work on robotics has been carried out to develop systems to help with in-field data acquisition to study the soil and crop spatial variability in sugarcane, soybean, and cotton [127,128]. However, these studies were conducted from an engineering perspective that was focused on design, navigation and control, rather than on integrating sensing technologies and deploying the robots to deliver value to farmers.

Technical and economic assessments of intelligent machinery applications in the context of Brazilian tropical agriculture are expected to expand in the coming years. With the unfavorable currency exchange rates expected in the next few years (USD/BRL ~ 5), we expect that more national technology will become competitive in many areas.

3.3.6. Remote Sensing and Unmanned Aerial Vehicles (UAVs)

Remote sensing is an essential tool that allows the spatial and temporal heterogeneity of large agricultural extensions to be monitored [129]. Globally, the first uses of remote sensing in PA started with a study conducted by Bhatti et al. [130] and have evolved in parallel with the advances in the spatial, temporal, and spectral resolution of systems and sensors [131]. In Brazil, the first study using remote sensing compared corn and soybean yield maps with the digital values of aerial photographs [132]. Despite the sensor's spectral limitations, the high spatial resolution obtained (resolution of 5 m) showed this

tool's potential. Subsequently, Almeida et al. [133] proposed an efficient methodology for predicting sugarcane yield that involved the analysis of historical yield data and spectral information in different bands of the Landsat-7 and ASTER systems. Despite these pioneering investigations, few studies have used free-access orbital imagery (e.g., Sentinel and CBERS) to characterize spatial variability for plant management, probably due to a lack of integration between the PA and remote sensing communities.

Remote sensing applications for soil characterization started with Demattê et al. [58]. These represent promising techniques for predicting textural attributes in tropical soils. Nevertheless, the importance of using methods that allow the identification of pixels within exposed soil (e.g., Shabou et al. [134]) is emphasized, seeking to avoid confusion between responses to soil and straw given the predominance of the no-tillage system in the country. Other studies have evaluated soil attributes through the spectral response of vegetation [135], indicating the possibility of using vegetation indices as auxiliary information for soil mapping.

In recent years, there have been increasing studies using UAVs, which are remote sensing platforms that have the convenience of allowing the attachment of sensors (e.g., cameras, LiDAR, etc.) and other equipment (e.g., sprayers, natural enemy dispersers, etc.). These studies are recent in Brazil and have been carried out using basic imaging sensors with different spectral resolutions. It is possible to highlight the use of hyperspectral cameras to monitor diseases in sugarcane and orange [113], biomass variability in soybean with a multispectral camera [96], and the use of 3D modeling to determine the biomass of sugarcane, oats, corn, and coffee [136–138]. Studies that have used imaging sensors to obtain vegetation indices have reported difficulties during the processing of the mosaics and atmospheric correction of the images due to the presence of shadows and light intensity variation in the image set. Therefore, despite the flexibility of UAVs in data collection and their high spatial resolution, their use still lacks a standardized processing structure for using these data [114,131].

In recent years, there has been a growing interest in technology in agronomic consultancies and large agricultural groups. Similarly, in the market scope, the offer of services for making orbital images of high spatial (at 3–5 m resolution) and temporal (at least three images per week) increased at costs ranging between USD 0.05 and USD 0.20 ha yr$^{-1}$. Thus, for the large-scale diffusion of remote sensing in Brazilian agriculture, we believe that efforts should be directed towards (i) studies indicating the economic benefits of the different approaches available; (ii) initiatives for technology transfer between scientists, producers, and data providers; and (iii) systems that allow automatic image processing and analysis tools to assess spatial and temporal variability in different soil and plant conditions.

### 3.3.7. Decision Support Tools

There were several studies focused on data analytics and modelling to turn field data into useful information for on-farm decisions. An approach based on the structuring and standardization of data as a basic principle for the better use of the information collected was presented by Murakami et al. [139]. Decision support tools have also been proposed for precision irrigation [140], the estimation of wheat yield [91], and corn yield using neural networks [141]; these approaches were used to anticipate possible factors limiting yield in each region of the field and allow in situ interventions. Santos and Saraiva [142] developed a reference model to delimit management zones, covering data collection, filtering, selection and grouping, and map evaluation. Driemeier et al. [143] created a computer system to support experimentation for sugarcane, seeking to integrate different data sources and facilitate the analysis of the relationships between them.

Despite research efforts in the development of various types of agro-computing tools in the PA context, it is noticeable that none of the approaches actually tackle decision making per se, i.e., they are normally focused on providing additional spatial information for the user, but the actual decision (for example, how much fertilizer to apply in each yield potential zone of a field) remains a significant challenge. Ultimately, farmers need to make

use of traditional agronomic frameworks (e.g., regional fertilizer charts) to come up with a site-specific decision, regardless of if such a framework is suitable or not at the site-specific level. Therefore, the development of digital decision tools based on aggregated data and experiments at the farm level to improve site-specific agronomic recommendations (e.g., Colaço et al. [144]) is a research field to be encouraged.

### 3.4. Sector Organization and Public Policies for Precision Agriculture in Brazil

To add context to Brazilian scientific production, below, we provide a description of how the academic, industry, and government communities organized themselves around the theme of PA in Brazil. PA sector organization has occurred in parallel with scientific research in Brazil. Universities and research centers started organizing academic events to present and discuss PA concepts and tools in the late 1990s. Industries and service providers related to PA also started to be structured in the first few years, as shown in the inventory made by Embrapa in 1999 [26]. As a result of the combination of some of these initiatives, the first ConBAP, a conference dedicated to PA issues was organized in 2004. Since then, the academic and business segments have acted together, which have allowed articulations with the Ministry of Agriculture, Livestock, and Supply (MAPA). In 2012, the Brazilian Commission of Precision and Digital Agriculture (CBAPD) was created as an advisory forum and was formed by entities representing the sector. However, PA service providers did not participate because they were not formally organized, and the formation of the Brazilian Association of Service Providers in PA (ABPSAP) was then promoted in 2015. Today, ABPSAP has more than 130 small and mid-sized companies as members, representing a significant part of the PA services in the country. The creation of the AsBraAP, a Brazilian association of PA, which happened in 2016, was promoted to bring the entire community together. People and companies form the AsBraAP, and, since 2018, it has been responsible for organizing the ConBAP, among other events. In addition, international events hosted in Brazil, such as South American Congress on PA (APSul América), have an important role in promoting PA among farmers, researchers, and other stakeholders from Brazil, Argentina, Uruguay, Chile, Paraguay, and Bolivia.

Due to the importance of PA in Brazil, in 2019, the federal government instituted the CBAPD, which acts under the MAPA, to promote the development of precision and digital agriculture in the country (decree no. 10052). The CBAPD's competencies are (i) to disseminate and promote the concept and techniques of PA; (ii) to disseminate the importance of PA for agricultural development and the promotion of socio-environmental sustainability; (iii) to support professional updating, training, and qualification programs and technical and scientific work related to PA; (iv) to generate and adapt affordable knowledge and technologies; (v) to propose public policies for the sector and ways of inserting PA in policies; (vi) to support the creation and updating of a public domain database of activities related to the sector; (vii) to implement and maintain a virtual discussion forum on PA; (viii) to identify structural demands and trends in PA in the country and abroad; and (ix) to promote coordination with public and private agents to define priority actions in the sector [145].

The recent movements related to digital contributions to agriculture have generated many new fronts for discussions, planning, and actions. At the same time, it has fomented expectations from farmers, with new challenges for the sector, which must be organized to meet the new demands. There is consensus in the community that digital solutions in PA must have a substantial impact on data generation and analysis to enable increasingly automated diagnostics and recommendations for the management of spatial variability and for the optimization of agriculture mechanization in a much more intense way than we have so far and without forgetting the small and mid-sized operations that have limited access to technology.

*3.5. Final Remarks*

Research in precision agriculture (PA) has evolved substantially over the last 25 years in Brazil. In parallel, the adoption of PA technologies has been spread out across the country, driven by research advances, sectorial organization, and investments by the public and private sectors. Our review described and discussed the trajectory of Brazilian PA research; the strengths, weaknesses, opportunities, and threats of the main researched areas are summarized in Table 1. Initially, research focused on recognizing and mapping the spatial variability of crop yield and soil properties; for a while, these were the two main pillars of PA in Brazil. Over time, PA research expanded, increasing the number of research groups, the number of publications, and the scientific cooperation among Brazilian and international researchers. With the advances in studies and collaboration, increased connectivity, and access to new technologies, other tools and areas of PA became emergent, such as the use of sensors to obtain high-resolution data of soil and plant attributes, the use of images from satellites and UAVs, and the introduction of decision support tools and systems (apps, software, cloud platforms, artificial intelligence among others).

While Brazilian scientific research may be considered to hold a leading position in the development of PA for tropical crops such as sugarcane, orange, and coffee, in most other instances, Brazil is mostly an importer of technology that was originally developed elsewhere. That is, scientific research and the field adoption of PA in Brazil is characterized by testing, using, and adapting imported technologies, which, in many cases, created barrier to the massive expansion of PA in the country. Currently, this scenario has been changing with the growth of well-trained human resources and advances in national industry, which should facilitate the integration and local development of technologies as well as the access of PA for farmers and service providers.

Another important issue in Brazilian PA research, which, in many respects, reflects a challenge for PA research worldwide, is the notorious emphasis given to tools and technologies for collecting site-specific digital data at the farm level and to site-specific input applications using agronomic decision frameworks that have actually been developed for traditional agronomy and that are thus outside the scope of site-specific management. For example, the vast majority (if not all) of variable-rate fertilizer application in Brazil, both in and outside of scientific studies, has relied on traditional fertilizer recommendation charts developed at the regional level and that were thus developed for average field and season conditions. Of course, none of these properly reflect the site-specific crop response to input applications, which goes heavily against PA principles. Therefore, it is imperative that research moves from a data collection focus towards new digitally based decision systems that are suitable for site-specific management. This issue was raised by Colaço et al. [144], for example. On-farm precision experimentation [146,147], a key enabler for such development, has been largely neglected in Brazilian research, especially for crops other than grains.

Therefore, key areas that remain largely unexplored and that should be encouraged in future research are (i) the development of digitally based decision tools that are underpinned by the systematized construction of large databases containing high-quality spatial and temporal information using multiple monitoring tools (e.g., soil/crop sensing and weather forecasting), field management and history information, and yield responses to applied input through on-farm precision experimentation; (ii) upscaling on-farm precision experimentation using well-established protocols within a comprehensive research network; (iii) integration between novel machine learning approaches and large databases to promote the implementation of digitally based decision support systems; and (iv) PA applications for environmental issues (e.g., C sequestration and reduction of pesticide usage).

**Table 1.** Strengths, weaknesses, opportunities, and threats (SWOT analysis) of each PA research area in Brazil that summarize the main messages and implications learned from this study.

| | Strengths | Weaknesses | Opportunities | Threats |
|---|---|---|---|---|
| Soil management | (i) Pioneer area with the largest number of studies and amount of knowledge generated over time; (ii) Large number of qualified researchers working in soil science; (iii) There are service providers and routine laboratories available across the country; (iv) Farmers recognize that optimized soil management is key for enhanced crop yields. | (i) Research has been restricted to studying the variability in chemical indicators, but little is known about physical and biological indicators; (ii) The infrastructure for the physical and biological sampling and analysis of soil are still incipient in most regions of the country; (iii) There are no public databases to share soil data collected by PA users. | (i) New sensors have recently become available for investigation (e.g., portable XRF); (ii) Possibility of knowledge integration with complementary areas (e.g., data science and biotechnology); (iii) The development of soil health assessment protocols to be used on a large scale; (iv) The use of PA principles to design protocols for soil carbon monitoring, reporting, and verification. | (i) Difficulty in generalizing robust agronomic algorithms for soil fertility characterization due to the high heterogeneity of soil composition in different areas; (ii) Loss of credibility in new technologies due to inappropriate commercial applications and/or excessive promises. |
| Phytosanitary management | More rational and environmentally responsible use of agrochemicals. | (i) Slow scientific advances over the last 25 years; (ii) Systematic sampling protocols are expensive, time-consuming, and tedious; (iii) Pests and diseases have dynamic behavior requiring temporal monitoring; (iv) Few scientists working with phytosanitary management in PA; (v) Incipient transference of scientific knowledge to market. | (i) New sensor systems that allow localized application of herbicides promoting more economical and sustainable strategies; (ii) Growing environmental awareness and public pressure to reduce pesticide load in agriculture. | (i) Discouragement of research investments due to difficulty in developing strategies for early detection of pests and diseases; (ii) Pesticide application schedule adopted to large-scale crops; (iii) The occurrence of new pests and diseases. |
| Remote Sensing/UAV | Large-scale mapping with low cost compared to traditional soil and plant sampling and analysis. | (i) High cloud density during the Brazilian crop season; (ii) Diagnostics are mostly late and do not allow interventions in the same crop season; (iii) Low applicability of the information to farmers; (iv) Only a few research groups working with technologies. | (i) Explore the potential of new UAV-compatible imaging techniques (e.g., thermal images); (ii) Potential for creating large databases with temporal information about agricultural crops and soil; (iii) Trend of reducing the spatial and temporal resolution of orbital satellites as well as reducing the price of images acquisition. | (i) Popularization of use without transferring the fundamentals behind the technique; (ii) Development of simplistic tools that overestimate the potential of the technique; (iii) External dependence on imaging and drone companies. |

**Table 1.** *Cont.*

| | Strengths | Weaknesses | Opportunities | Threats |
|---|---|---|---|---|
| Decision support tools | Data analytics, including protocols and modelling, for various purposes have been developed to provide high-quality spatial information for improved agronomic decisions. | On-farm decisions remain reliant on traditional agronomic frameworks that are not suitable for site-specific management. | Development of on-farm experimentation methods and protocols for the generation of large on-farm digital databases. | Difficulty in ensuring data quality for the construction of large datasets and data privacy issues. |
| Plant management | (i) Large scientific background built over 25 years; (ii) Large number of researchers working on this topic; (iii) Diagnosis of the spatio-temporal variability in crop yield; (iv) Protocol to adjust crop demands with environmental conditions (plant populations and fertilization). | (i) Limited and low adherence to yield monitors and little use of yield maps for decision-making; (ii) High cost and consequent low adoption of real-time sensor-based fertilization management. | (i) Optimize decision making via integration of sensors and the application of on-farm trials; (ii) Growing food, feed, fiber, and biofuel demands will push Brazilian farmers to be more efficient and productive; (iii) Yield monitoring is becoming cheaper and popular; (iv) Evolution in data sharing, storing, and processing capacity. | (i) External dependency of most plant sensors and devices. |
| Machinery, equipment, and autonomous vehicles | (i) Establishment of ISOBUS as an open standard for the interconnection of electronic systems; (ii) High engagement of the national industry. | (i) Advances in this area are developed and guided by the companies in this sector, with minimal involvement from academia; (ii) The research is mostly focused on testing and adapting existing technologies than developing new ones in this field. | (i) New crop/soil monitoring strategies using technology embedded in agricultural machines; (ii) Evolution of robotics applied to agriculture. | |
| GNSS | (i) Worldwide system with 24 h operation and accuracy compatible with agricultural operations; (ii) GNSS embedded in the machines increase the efficiency of agriculture operations (sowing, spraying, CTF, etc.). | (i) Necessity of monitoring ionospheric scintillation phenomenon and stop operations at peak times; (ii) Low accuracy of navigation GPS devices used in soil sampling and other activities. | (i) Improvements in communication signal technology and multi-constellation receivers have the potential to bring better performance at a more affordable price; (ii) New smartphones and free applications with GNSS signal with suitable accuracy for most agricultural activities. | Dependency on systems from foreign countries. |

Finally, public and private investments in research and extension are fundamental to collectively promote PA in Brazil. We believe that PA technologies are and will continue to be critical in making Brazilian agriculture more productive, efficient, competitive, and sustainable in the coming decades. This study revealed insights that can help identify research and market opportunities for PA actors, including farmers, researchers, public and private companies, and policymakers.

**Supplementary Materials:** The following supporting information can be downloaded at https://www.mdpi.com/article/10.3390/agriculture12111882/s1: Table S1: List of articles gathered in the bibliometric review; and in Table S2: List of main textbooks related to precision agriculture published in Brazil.

**Author Contributions:** Conceptualization, M.R.C.; methodology, M.R.C., M.T.E. and R.G.T.; validation, T.R.T., R.Y.I. and J.P.M.; formal analysis, M.R.C., M.T.E., R.G.T. and J.M.D.; data curation, M.R.C., M.T.E., R.G.T., J.M.D. and M.M.; writing—original draft preparation, M.R.C., M.T.E., T.R.T., R.G.T., J.M.D., M.M. and O.H.d.C.P.; writing—review and editing, A.F.C., R.Y.I. and J.P.M.; visualization, A.F.C., R.Y.I. and J.P.M.; supervision, M.R.C.; funding acquisition, M.R.C. All authors have read and agreed to the published version of the manuscript.

**Funding:** T.R.T. was funded by the São Paulo Research Foundation (FAPESP), grant number 2020/16670-9.

**Institutional Review Board Statement:** Not applicable.

**Data Availability Statement:** Not applicable.

**Acknowledgments:** M.R.C. thanks the CNPq for his Research Productivity Fellowship (311787/2021-5).

**Conflicts of Interest:** The authors declare no conflict of interest.

## Appendix A

Figure A1 shows the evolution (from 1996 to 2020) of scientific publications regarding precision agriculture (PA) in relation to scientific publications within the major area of agriculture, which was surveyed using Scopus with the following query: "(TITLE-ABS-KEY (Brazil)) AND (LIMIT-TO (SUBJAREA, "AGRI")) AND (LIMIT-TO (PUBYEAR, YYYY))".

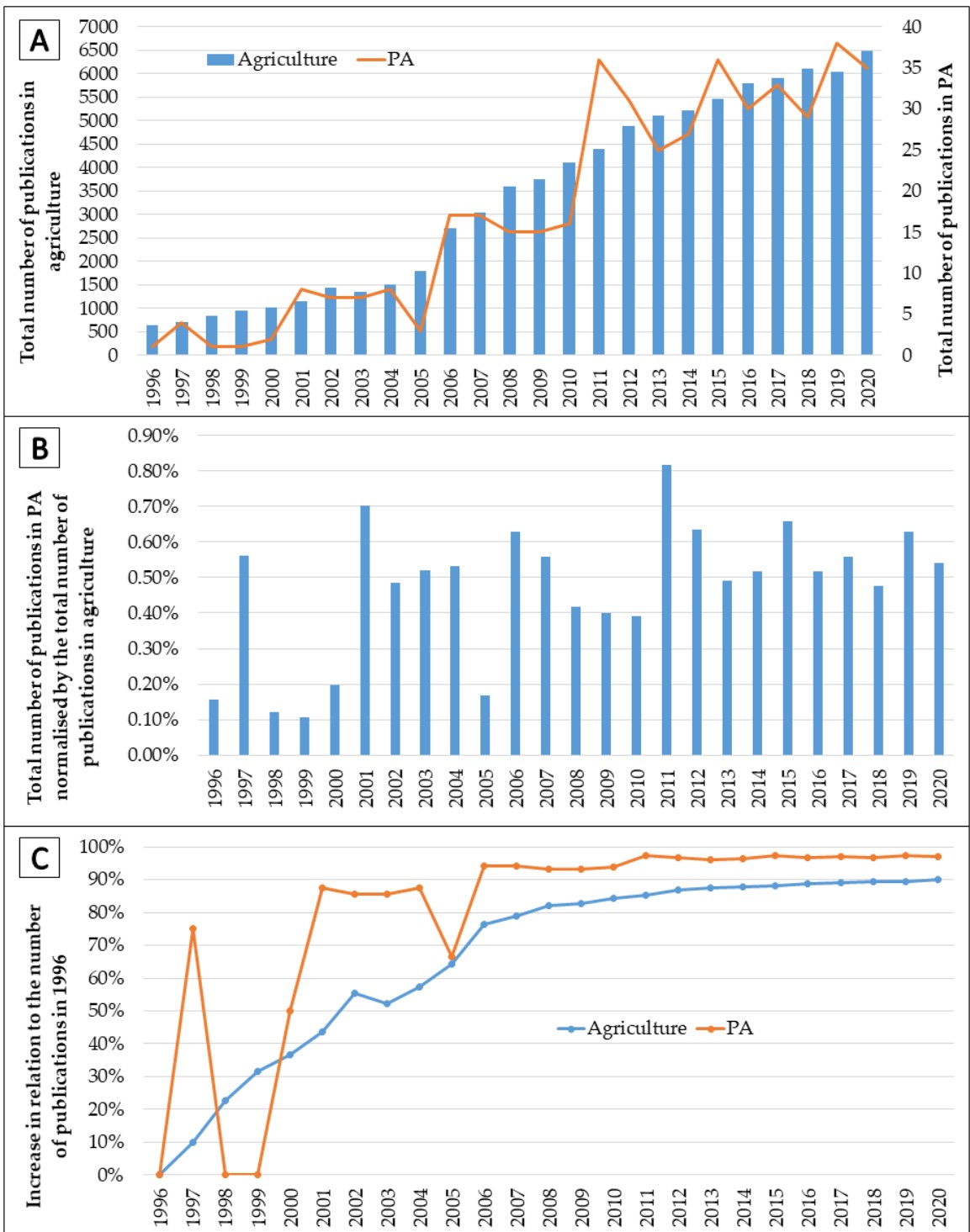

**Figure A1.** Evolution (from 1996 to 2020) of the total number of publications related to agriculture (**A**); evolution (from 1996 to 2020) of the total number of publications related to precision agriculture (PA) normalized by the total number of publications in agriculture (**B**); and increase in the percentage of the number of publications (in PA and agriculture) in relation to the year 1996 (**C**), the year of the first publication on PA in Brazil.

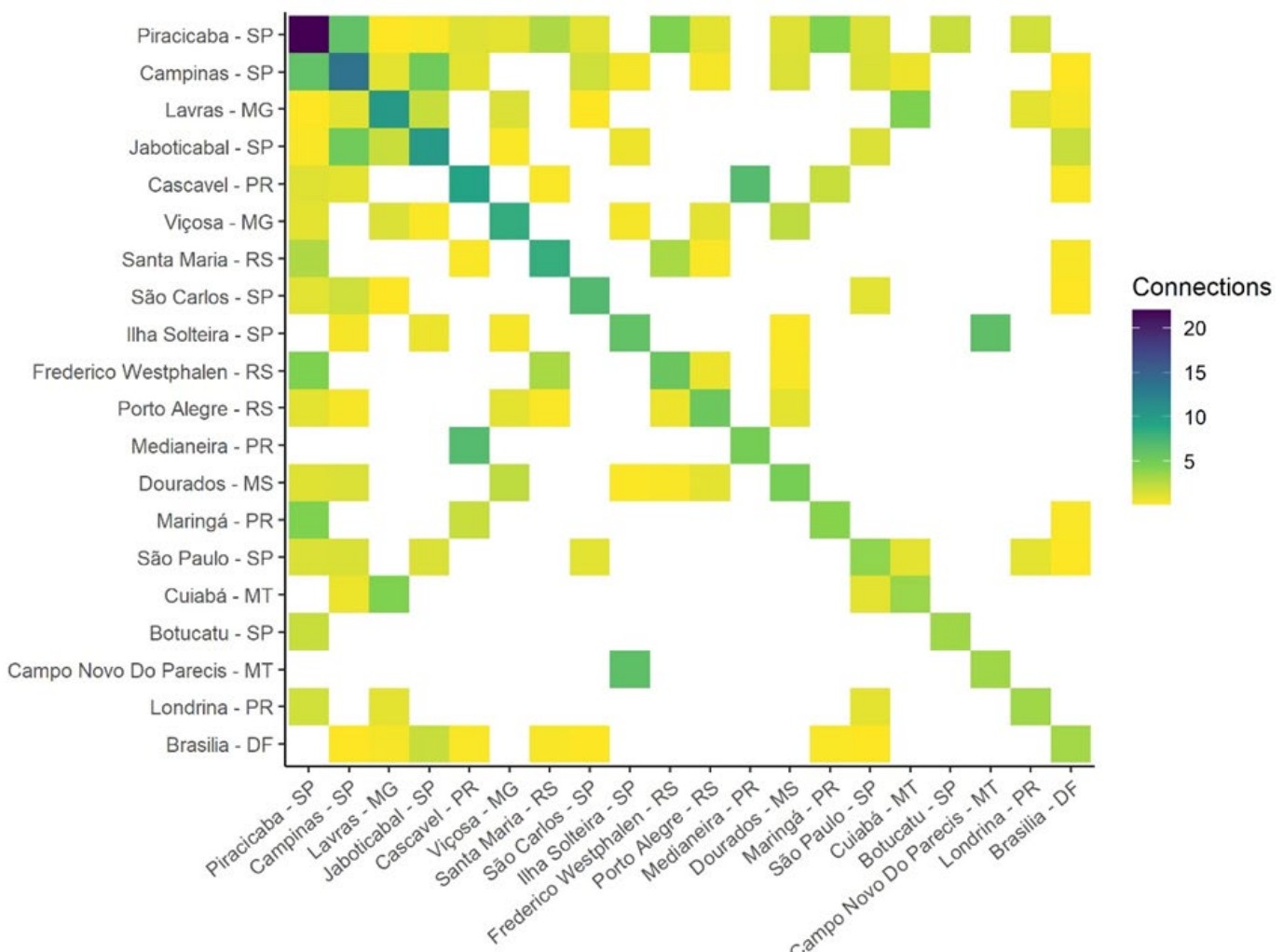

**Figure A2.** Matrix with the top 20 cities with the highest number of interactions with other research groups that resulted in scientific papers; on the main diagonal, the sum of the total weight of connections for each city is shown, and the other cells show the weight of each pair of cities.

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
