# Peer review of "Precision Agriculture in Brazil: The Trajectory of 25 Years of Scientific Research"

_agriculture, doi:10.3390/agriculture12111882_

Round 1

Reviewer 1 Report

The work presented in this paper focuses on a detailed review of agricultural evolution in Brazil during 25 years. Please find the various remarks and recommendations necessary to improve the quality of the final version of this paper.

Minor typos

1. In line 43 the reference 1 must be in the end of the sentence as follows « Food demand should double from 2005 to 2050 due to the global population'sgrowth [1]. »

2. Between line 54 and 56 the authors defined agriculture as follows: « Precision agriculture comprises a set of technologies that combine sensors, information and positioning systems and modern machinery to manage the spatial-temporal variability of production factors and optimize agricultural yields ». Please change it as follows:“Precision agriculture comprises a set of technologies that combine sensors, information, systems positioning, and modern machinery based on data processing to manage the spatial-temporal variability of production factors and optimize agricultural yields.”

3. In line 60 change braod to general

4. In line 52 and 63 merge the two paragraphs because they are referring to the same thing

5. Line 60: The abbreviations should be written open in the first place. Please write the long version of PA.

6. In line 82, you have to move the references [20, 21] to the end of the sentence line 84. And the same remark for line 92.

7. In the line 112 it is better to make a return to line.

8. All the terms and acronyms need to be defined the first time they appear.

Commentes and questions

1. I see that the introduction presentation is very general, I recommend to the authors to focus on one or more issues to be addressed please see the reference and try to rewrite the introduction  “https://doi.org/10.1080/09064710.2021.2024874”.

2. The abstract should provide the main outputs, such as which problem requires which technologies or what the recommended solutions to increase the productivity of agricultural fields.

3. A chart showing the systematization of the manuscript may help to understand the structure of the manuscript. This helps to simplify and improve the structure of the text.

4. I suggest adding this definition of precision agriculture (PA): https://www.ispag.org/about/definition.

5. It is necessary to add a detailed explanation below the figure 1 in order to be comprehensible.

6. Lines 103-108: How these statements are supported?

7. In the material and method part it is recommended to change the title of the section as it does not reflect the content of this section.

8. In figure 3 line 202, you have to explain the abbreviations RR, AC, MT, MS……..

9. In figure 4 the lines in this figure are based on which study and it is necessary to justify the link between the countries to support the results obtained in this figure.

10. In the sections 3.3.2 and 3.3.3, please merge these two sections because plant management and soil management have the same objective.

11. In section 3.3.6, An important point to be mentioned is that UAV are platforms that are used to attach sensors (cameras), and these sensors are used to generate data from the agricultural fields. It would be interesting to describe the sensor instead of the UAV. The sensor is what generates the data that is monitored in the field.

12. A table with the type of algorithm, type of images used, and their application can be added to 3.3.6 section line 586. Also, inform in which crops they have been used.

13. In table 1 line 748, a description of which application in agriculture is missing.

14. In the precision agriculture literature, several works have been published that are based on complete systems based on convolutional algorithms or artificial intelligence. What I recommend to the authors of the paper is to add a section based on the different sensors and stations processing the data collected by the different sensors.

15. As the authors have shown 25 years of experience feedback, I recommend adding a section towards the end of the results section that focuses on the authors viewpoint on agriculture after 25 years.

Reviewer 2 Report

Queries/suggestions/observations are included in the attachment. 
